# Association between Maternal Anxiety and Children’s Problem Behaviors: A Systematic Review and Meta-Analysis

**DOI:** 10.3390/ijerph191711106

**Published:** 2022-09-05

**Authors:** Zhanmei Song, Jie Huang, Tianqi Qiao, Jingfeng Yan, Xueying Zhang, Dengcheng Lu

**Affiliations:** 1Institute of International Education, Wenzhou University, Wenzhou 325035, China; 2School of Education, Wenzhou University, Wenzhou 325035, China

**Keywords:** maternal anxiety, problem behavior, meta-analysis, moderating effect, preschoolers

## Abstract

Although numerous studies have found that maternal anxiety is a risk factor for the development of children’s problem behaviors, and there is a possible role of genes in the association between the two. And anxious mothers caring for their children can also affect the development of children’s problem behaviors. However, there is also considerable evidence from studies that refute this view. This study used a meta-analysis to explore the relationship between maternal anxiety and preschool children’s problem behaviors. Through literature retrieval and selection, in terms of the criteria for inclusion in the meta-analysis, 88 independent effect sizes (34 studies, 295,032 participants) were picked out as meta-analysis units. The test for heterogeneity illustrated that there was significant heterogeneity in 88 independent effect sizes, while the random effects model was an appropriate model for the subsequent meta-analysis. The publication bias test indicated that the impact of publication bias was modest but the major findings remained valid. In addition, in terms of the tentative review analysis and research hypotheses, the random effects model was used as a meta-analysis model. The research revealed that maternal anxiety was significantly positively correlated with preschool children’s internalizing problem behaviors, externalizing problem behaviors, and overall problem behaviors. The moderating effect analysis showed that region and gender of the child affected the relationship between maternal anxiety and children’s internalizing problem behaviors and externalizing problem behaviors, and region, child’s age and gender, mother’s age, and education level affected maternal anxiety and preschool children’s problems behavioral relationship. Hence, these results affirmed the role of maternal anxiety and emphasized the need to pay attention to the demographic characteristics and cultural background of the subjects during the research process and consider the generalizability of the conclusions under different circumstances.

## 1. Introduction

In the new era of the 21st century, the concept of education is changing, and the family structure and parenting style are changing at the same time. Although the material conditions and infrastructure have been improved, how to raise an excellent child has now become a new focus. Parenting responsibilities and pressures force parents to spend a lot of time and energy caring for and nurturing their children. In addition, the shortage of educational resources, the comparison between parents and peers, and the lack of parenting experience virtually increase their tension and anxiety.

From the perspective of family education and parenting methods in various countries, mothers are the most important caregivers in early childhood, for they have the longest contact with children and can exert the greatest influence on them. Studies have found that there is a correlation between mothers’ emotions and children’s emotions. However, previous studies have mainly explored the causes of mothers’ anxiety, mothers’ mental health status, and the direct or indirect effects of changes in parenting styles on children. For example, the results of a recent follow-up study showed that at different ages of children, there is a significant direct correlation between maternal anxiety and children’s internalizing problem behaviors, but there is no significant correlation between maternal anxiety and children’s externalizing problem behaviors [1]. Meanwhile, a cross-sectional study in Asia showed that maternal psychological control plays a significant mediating role between maternal anxiety and children’s problem behaviors [2]. There is a less in-depth exploration of the relationship between maternal anxiety and young children’s problem behaviors. Rees conducted a relatively systematic meta-analysis of related literature between 2002 and 2017 [3]. However, because the meta-analysis did not include the latest research results in the past ten years, and the quality of the literature included in the study was uneven, the reference to the meta-analysis results was limited. If we can analyze whether and in what way maternal anxiety can affect young children’s problem behaviors through further meta-analysis, and analyze the moderating effects of different characteristics, we can also expand the topics of the impact of maternal anxiety on children’s development. Additionally, we hope that the research content will point out the direction for future research. It is expected that in the future, more effective programs will be put forward to improve the impact of maternal anxiety symptoms on children’s development to ensure the development of children.

### 1.1. Maternal Anxiety

Anxiety is a complex but individual emotional response. Different scholars have different views on the definition of the specific concept of anxiety. Some scholars believe that anxiety is unique to people, while others believe that there are different types of anxiety. For example, Chinese scholar Bowei Luo defined anxiety as an emotional state in which self-esteem and self-confidence are frustrated, and feelings of failure and guilt, nervousness, and a sense of fear are increased due to the threat of inability to achieve goals or inability to overcome obstacles [4].

In terms of the development of children, maternal anxiety mainly includes a series of nervousness and oversensitivity to children’s diet, growth, development, behavior, appearance, clothing, temperament, peer, and parent-child relationships [5]. Compared with the normal sensitivity and responsiveness to children’s needs, anxious mothers are apprehensive about their children’s physical and mental development, fearing that their children will develop poorly or fall behind. In addition, in order to obtain a sense of parenting security, parents also strictly limit their children’s exploratory behavior. There are even some mothers who are always worried about the growth and development of their children. Over time, they will suffer from loss of appetite, poor sleep, nervousness, and restlessness. In the end, their maternal anxiety will fall into a vicious cycle.

### 1.2. Problem Behaviors of Preschool Children

Problem behaviors can also be called behavioral problems. In the 1920s, after Wickman researched problem behaviors, this field attracted the attention of domestic and foreign researchers. Up until now, fairly rich research results have been accumulated. However, there was no consensus on the exact definition of problem behaviors, which may be due to the different cultural backgrounds, positions, and research methods of domestic and foreign scholars. Wickman, an American scholar, believed that problem behaviors refer to the conflict between individual behavior and social behavioral norms. Furthermore, some scholars believe that problem behaviors are all problems that make people hard adapt to [6], and are a series of behaviors that are contrary to the behavioral standards, produced by children in the process of socialization [7]. Problem behaviors include emotion, attention deficit, and peer relationship problem behaviors [8].

The definition of problem behaviors by Chinese scholars tends to have the following viewpoints: all abnormal behaviors that hinder individual and social development are problem behaviors, including externalizing and internalizing problem behaviors [9]. In addition, Liping Chi and Ziqiang Xin believed that a series of behaviors, such as non-compliance with social norms, out of control emotional management, and ideology and morality against the norm, are all problem behaviors [10]. No matter which point of view is used to define problem behaviors, it is all discussed around the difficult adaptation behaviors that children and adolescents produce in the process of socialization.

### 1.3. The Relationship between Maternal Anxiety and Preschool Children’s Problem Behaviors

Based on social learning theory, maternal anxiety can influence children’s anxiety levels through genetics and parenting style [11]. Gete conducted a longitudinal tracking study on 396 mother–child pairs and found that maternal anxiety can be directly transmitted to children, which in turn has an impact on child development [12]. On this basis, Allen reviewed the literature and proposed that genetic susceptibility, environmental stress, parenting sensitivity, parental pressures, parenting experience, and the interaction among these factors can be used as the explanatory factors for the two significant characteristics of trait anxiety, intergenerational inheritance, and bidirectional regulation [13]. Furthermore, Cartwright-Hatton found that environmental factors were more likely to explain the intergenerational inheritance of anxiety than genetic factors [14]. Chinese scholar Lin studied 71 preschool children and found that preschool children’s emotional problems such as anxiety and depression were significantly correlated with their mothers’ emotional problems [15].

Some researchers believe that parental mood directly affects children’s behavioral problems. For instance, Maoz and others found that parents with high mood fluctuation were more likely to overestimate internalized and externalized behavioral problems of preschool or school-age children [16]. If mothers are in a negative mood such as with anxiety before or after childbirth, their children are more likely to have behavior problems during the preschool period [17]. Many children who have been exposed to maternal anxiety for a long time also show some similar symptoms of anxiety [18]. Moreover, living with anxious parents for a long time can also adversely affect children’s mental health and academic performance [19]. Other researchers, however, argue that parents’ moods are not directly related to children’s behavioral problems, but parents will affect children’s behavior problems through intermediate factors such as parenting behaviors and parent–child relationships. For example, parents in a negative mood state tend to show indifference, hyposensitivity, lack of patience, and other behaviors in parent–child interaction and do not show more joy in reuniting with children after separation [20]. Therefore, on the one hand, negative moods can weaken parents’ positive parenting behaviors. Compared with non-anxious mothers, parents in a long-term negative mood are more likely to evaluate their children negatively [21]. They are less involved in interactions with their children, and children are more likely to have negative moods and a decrease in activity [20]. On the other hand, a negative mood can lead to family dysfunction. Studies have found that parents in a negative mood for a long time have less emotional exchange and communication in the family, making the whole family in a depressing or conflicting atmosphere, which in turn affects children’s social behavior and results in more problem behaviors in children.

However, other studies have found that maternal anxiety and depressive symptoms are significantly correlated with externalizing problem behaviors in girls, but are not associated with externalizing problem behaviors in boys [22]. At the same time, some scholars found that there was a negative correlation between maternal anxiety and the number of children’s health problems and temperament inhibition, but it was not associated with children’s internalizing problem behaviors, such as child anxiety [23]. Results of a long-term follow-up study showed that mothers experienced anxiety in their children at 3 months, 6 months, and 12 months, 10 years, and 12.5 years. Mother-reported child internalizing problem behaviors were significantly associated with maternal anxiety, but not for children of externalizing problem behaviors, and were not significantly associated with maternal anxiety or depressive symptoms [1]. As for the relationship between maternal anxiety and children’s externalizing problem behaviors, some scholars have also conducted related research. For example, a study by Zlomke found no correlation between maternal anxiety and children’s externalizing problem behaviors [24]. Although many empirical studies have proved that there is a significant correlation between maternal anxiety and children’s problem behaviors, as shown in the above studies, the correlation between these two is not clear, and some studies even have opposite conclusions that these two are not related. Because of this, Rees conducted a more systematic meta-analysis of the relationship between maternal anxiety and children’s internalizing problem behaviors [3]. However, because the study only included a total of 14 studies from 2002 to 2017, the number of studies was small and the quality was uneven, which severely limited the ability of this meta-analysis to draw clear conclusions. Besides, some of the included studies in this meta-analysis had problems such as lack of data and exclusion criteria, limited geographic coverage, and loss of researchers, which also meant that the generalizability of the meta-analysis results may be limited. Moreover, the focus of this meta-analysis was to distinguish the effects of maternal anxiety symptoms at different periods on children’s emotional problems, in which the age range of the children tested was very broad, which meant that the study could not focus on the effects of maternal anxiety on preschool children’s emotional problems. Finally, the meta-analysis concluded that maternal prenatal anxiety, pregnancy anxiety, and postpartum anxiety all showed a small effect on children’s emotional problems, and did not analyze the moderating effects of other factors in detail. Considering the limitations of many problems in this study to obtain valid conclusions, the true effect size of the relationship between maternal anxiety and preschool children’s problem behaviors, and what factors affect the effect size, the reference to this research question was extremely limited.

## 2. Materials and Methods

### 2.1. Literature Search

First, two reviewers respectively conducted a literature search in both Chinese and English. The search time period was set from the establishment of the database to 1 July 2022, and all the relevant literature including the latest literature was retrieved as far as possible. The search process was as follows: First, articles were retrieved from the CNKI Database, the China Science and Technology Journal Database, the Wanfang Database, the China Master’s Theses Full-text Database, and the China Doctoral Dissertations Full-text Database. Second, articles were retrieved from EBSCO, JSTOR, SAGE, Springer, Elsevier, and ProQuest Dissertation and Theses databases. We used the keywords ‘maternal anxiety, parenting anxiety’, ‘children, child’ and ‘problem behavior, externalizing problem behavior, internalizing problems, emotional problems, inattention, conduct problem, peer relationship, aggression, aggressive behavior, disciplinary offense, delinquent behavior’. At the same time, we also searched in Google Scholar. Any disagreements that arose were jointly resolved by a third reviewer for a final consensus.

### 2.2. Criteria for Literature Inclusion

For the relevant studies searched, we decided whether to include them in the subsequent meta-analysis according to the following criteria: (a) empirical studies with numerical results must be reported, while purely theoretical and review studies are excluded; (b) studies must examine the relationship between maternal anxiety and preschool children’s problem behaviors or any other indicators; (c) if the same sample data is used and published by different independent institutes, the literature providing more detailed sample information will be selected for coding; and (d) studies that did not report a full effect size are excluded. Finally, we obtained 34 papers that met the requirements for the meta-analysis. Among them, there were 26 studies in English and 8 studies in Chinese.

The flow chart of document retrieval is shown in Figure 1. A total of 3747 articles were found through a computer search and a manual search. After filtering out repeated articles and reviews, there were 2832 articles. Then, the titles and abstracts were manually searched, 2717 articles were excluded, and the remaining 115 full-text papers were screened. As a result, 81 articles were excluded, of which 46 were inconsistent in research methods and 35 were inconsistent in outcome indicators.

### 2.3. Coding

All articles were independently screened by two reviewers for the title, abstract, and full text based on predefined criteria, and disagreements were resolved by a third reviewer. The collected articles were coded for the following features: first author, year, region, sample size, children’s age, gender and its proportion, maternal age, maternal education level, follow-up time, variable types, and outcome indicators. In addition, for studies for which data couldn’t be extracted, we obtained data by contacting the authors, see Table 1.

For each independent sample, an effect size was obtained. At the same time, we took into consideration that some studies focused on multiple behavioral variables (internalizing problem behaviors and externalizing problem behaviors), reported the results of different sample groups (male and female), and reported the results of multiple measurements (follow-up study). We presented the multiple independent effect sizes contained in each study’s literature. Therefore, some research documents would contain multiple independent effect sizes. In the end, we had a total of 88 independent effect sizes.

### 2.4. Meta-Analytic Process

#### 2.4.1. Effect Sizes

The data of this study were analyzed by the comprehensive meta-analysis software CMA2.2 and selected coefficient (r) as effect size. CMA2.2 converted various other effect values reported in the original independent study, such as the t-value, and Chi-square value, into correlation coefficients (r). If only the correlation coefficients of each dimension were reported, the correlation coefficient synthesis method (r-Fisher Z) was used to obtain the correlation coefficient between maternal anxiety and children’s problem behaviors in the total sample of subjects [25] for later analysis.

#### 2.4.2. Model Selection

Meta-analysis generally adopts a fixed effects model and a random effect model. The fixed effects models assume that the true effect sizes of each independent study included in the meta-analysis are the same, and that differences in results between studies are caused by random error. A random effects model assumes that each independent study included in the meta-analysis has its true effect value, and the differences in results between different studies are caused by different true effect values and random errors [26].

Specifically, if the average effect value of the meta-analysis is only for the population included in the meta-analysis, and does not extend to other populations, a fixed effects model should be used. However, when the age of the subjects and the measurement indicators are different in independent studies, a random effects model should be used because these differences will have an impact on the final result [27]. In addition to exploring the main effect of maternal anxiety on problem behaviors, this study also explored possible moderating variables between these two, so a random effects model was more appropriate for this study. In the subsequent meta-analysis, heterogeneity tests were used to validate our model selection.

#### 2.4.3. Publication Bias

Publication bias is considered to have occurred when the published research literature fails to represent the population of the research that has been completed in the field [28]. The result of publication bias is the incompleteness of the research literature in a particular field, which can seriously affect the results of the meta-analysis. Any meta-analysis should be concerned with publication bias [29]. To solve the problem of publication bias, we first obtained unpublished literature as much as possible in the literature search stage. In the subsequent meta-analysis process, we would use funnel plots, Rosenthal’s Fail-safe N, Egger’s test, and other methods to evaluate the publication bias of this meta-analysis.

**Table 1 ijerph-19-11106-t001:** Characteristics of the studies included in the study.

Main Author	Year	Region	Sample Size	Age of Children	Gender (Male %)	Maternal Age	Maternal Education	Period	Variable Type	Correlation Coefficient
Ali [29]	2019	Americas	182	T	<50%	30~35	60~80%	N	E	0.300
Ali [29]	2019	Americas	182	T	<50%	30~35	60~80%	N	I	0.350
Arikan [30]	2021	Asia	537	T	50~65%	30~35	<60%	N	I	0.510
Arikan [30]	2021	Asia	537	T	50~65%	30~35	<60%	N	E	0.500
Antúnez [22]	2018	Europe	311	T	<50%	Not reported	Not reported	N	P	0.130
Bangirana [31]	2021	Africa	116	T	<50%	Not reported	<60%	N	P	0.163
Clavarino [32]	2009	Australia	428	C	50~65%	<30	>80%	N	E	0.220
Clavarino [32]	2009	Australia	400	C	50~65%	<30	>80%	N	E	0.238
Clavarino [32]	2009	Australia	365	C	50~65%	<30	>80%	N	E	0.261
Clavarino [32]	2009	Australia	608	C	50~65%	30~35	>80%	N	E	0.204
Clavarino [32]	2009	Australia	298	C	50~65%	30~35	>80%	N	E	0.237
Clavarino [32]	2009	Australia	387	C	50~65%	30~35	>80%	N	E	0.256
Clavarino [32]	2009	Australia	332	C	50~65%	30~35	>80%	N	E	0.278
Ning [33]	2021	Asia	12657	Not reported	50~65%	Not reported	Not reported	N	P	0.112
Di Stefano [34]	2014	Americas	993	Not reported	50~65%	Not reported	Not reported	N	E	0.120
Di Stefano [34]	2014	Americas	993	Not reported	50~65%	Not reported	Not reported	N	I	0.270
Frigerio [35]	2021	Europe	90	T	50~65%	30~35	Not reported	N	I	0.470
Frigerio [35]	2021	Europe	90	T	50~65%	30~35	Not reported	N	I	0.520
Frigerio [35]	2021	Europe	90	T	50~65%	30~35	Not reported	N	I	0.380
Frigerio [35]	2021	Europe	90	T	50~65%	30~35	Not reported	N	I	0.460
Frigerio [35]	2021	Europe	90	T	50~65%	30~35	Not reported	N	E	0.250
Frigerio [35]	2021	Europe	90	T	50~65%	30~35	Not reported	N	E	0.310
Frigerio [35]	2021	Europe	90	T	50~65%	30~35	Not reported	N	E	0.330
Frigerio [35]	2021	Europe	90	T	50~65%	30~35	Not reported	N	E	0.400
Frigerio [36]	2022	Europe	94	T	50~65%	30~35	>80%	E	I	0.275
Frigerio [36]	2022	Europe	88	T	50~65%	>35	>80%	E	I	0.115
Frigerio [36]	2022	Europe	59	C	50~65%	>35	>80%	E	I	0.480
Frigerio [36]	2022	Europe	94	T	50~65%	30~35	>80%	E	E	0.285
Frigerio [36]	2022	Europe	88	T	50~65%	>35	>80%	E	E	0.130
Frigerio [36]	2022	Europe	59	C	50~65%	>35	>80%	E	E	0.365
Gagne [37]	2019	Americas	198	T	50~65%	Not reported	Not reported	N	E	0.160
Gagne [37]	2019	Americas	198	T	50~65%	Not reported	Not reported	N	E	0.230
Gagne [37]	2019	Americas	1994	C	Not reported	30~35	>80%	N	I	0.220
Gagne [37]	2019	Americas	1994	C	Not reported	30~35	>80%	N	E	0.230
Zilin [15]	2019	Asia	71	C	>65%	Not reported	Not reported	N	P	0.544
Gjerde [38]	2020	Europe	40457	T	50~65%	Not reported	Not reported	N	I	0.150
Gjerde [38]	2020	Europe	40457	T	50~65%	Not reported	Not reported	N	I	0.140
Gjerde [38]	2020	Europe	40457	T	50~65%	Not reported	Not reported	N	I	0.170
Gjerde [38]	2020	Europe	40457	T	50~65%	Not reported	Not reported	N	E	0.110
Gjerde [38]	2020	Europe	40457	T	50~65%	Not reported	Not reported	N	E	0.110
Gjerde [38]	2020	Europe	40457	T	50~65%	Not reported	Not reported	N	E	0.150
HanetzGamliel [39]	2021	Asia	141	C	<50%	>35	<60%	E	E	0.260
HanetzGamliel [39]	2021	Asia	141	C	<50%	>35	<60%	E	I	0.370
Hautmann [40]	2014	Europe	106	C	>65%	30~35	Not reported	N	E	0.340
Hautmann [40]	2014	Europe	106	C	>65%	30~35	Not reported	N	E	0.360
Hautmann [40]	2014	Europe	106	C	>65%	30~35	Not reported	N	E	0.400
Loomans [41]	2011	Europe	3446	C	50~65%	30~35	<60%	N	P	0.280
Marceau [42]	2015	Americas	308	Not reported	50~65%	Not reported	Not reported	N	E	0.020
Marceau [42]	2015	Americas	308	Not reported	50~65%	Not reported	Not reported	N	E	0.080
Marceau [42]	2015	Americas	308	Not reported	50~65%	Not reported	Not reported	N	I	-0.003
Marceau [42]	2015	Americas	308	Not reported	50~65%	Not reported	Not reported	N	I	0.080
McFarlane [43]	2014	Americas	300	C	Not reported	30~35	Not reported	N	I	0.611
McFarlane [43]	2014	Americas	300	C	Not reported	30~35	Not reported	N	E	0.494
McFarlane [43]	2014	Americas	300	C	Not reported	30~35	Not reported	N	P	0.666
Mount [23]	2010	Americas	83	T	>65%	30~35	Not reported	N	I	0.110
Porter [44]	2019	Australia	191	T	Not reported	<30	60~80%	N	P	0.300
Porter [44]	2019	Australia	191	T	Not reported	<30	60~80%	N	P	0.180
Porter [44]	2019	Australia	191	T	Not reported	30~35	60~80%	N	P	0.280
Porter [44]	2019	Australia	191	T	Not reported	30~35	60~80%	N	P	0.350
Ying [2]	2019	Asia	335	C	50~65%	30~35	<60%	N	E	0.320
Ying [2]	2019	Asia	335	C	50~65%	30~35	<60%	N	E	0.140
Huihui [45]	2016	Asia	3653	Not reported	50~65%	Not reported	<60%	N	P	0.078
Huihui [45]	2016	Asia	3375	Not reported	50~65%	Not reported	<60%	N	P	0.091
Teramoto [46]	2005	Asia	670	T	Not reported	Not reported	Not reported	N	P	0.400
Teramoto [46]	2005	Asia	670	T	Not reported	Not reported	Not reported	N	E	0.400
Teramoto [46]	2005	Asia	670	T	Not reported	Not reported	Not reported	N	I	0.330
Thorsteinsdottir [47]	2018	Europe	623	C	<50%	>35	60~80%	N	P	0.562
Tichelman [48]	2021	Europe	1528	Not reported	Not reported	30~35	<60%	N	E	0.247
Tichelman [48]	2021	Europe	1528	Not reported	Not reported	30~35	<60%	N	I	0.214
wai [49]	2017	Asia	61	T	<50%	30~35	Not reported	N	I	0.670
wai [49]	2017	Asia	61	T	<50%	30~35	Not reported	N	E	0.520
Yueyuan [50]	2020	Asia	506	C	50~65%	Not reported	<60%	N	I	0.208
Wiggins [51]	2019	Americas	672	Not reported	>65%	30~35	>80%	N	P	0.127
Yoldaş [52]	2019	Asia	40	T	<50%	30~35	<60%	N	I	0.430
Yoldaş [52]	2019	Asia	40	T	<50%	30~35	<60%	N	E	0.512
Yoldaş [52]	2019	Asia	40	T	<50%	30~35	<60%	N	P	0.488
Yoldaş [52]	2019	Asia	40	T	<50%	30~35	60~80%	N	I	0.412
Yoldaş [52]	2019	Asia	40	T	<50%	30~35	60~80%	N	I	0.403
Yurdusen [53]	2013	Asia	204	C	50~65%	30~35	60~80%	N	I	0.420
Yurdusen [53]	2013	Asia	204	C	50~65%	30~35	60~80%	N	E	0.330
Yurdusen [53]	2013	Asia	204	C	50~65%	30~35	60~80%	N	P	0.430
Yan [54]	2011	Asia	308	Not reported	<50%	Not reported	60~80%	N	I	0.418
Zhang [55]	2022	Americas	831	T	50~65%	<30	Not reported	N	P	0.280
Hao [56]	2021	Asia	3097	Not reported	50~65%	Not reported	<60%	E	P	0.278
Yang [57]	2014	Asia	355	Not reported	50~65%	Not reported	60~80%	N	I	0.351
Zlomke [24]	2020	Americas	64	C	>65%	30~35	60~80%	N	P	0.045
Zlomke [24]	2020	Americas	64	C	>65%	30~35	60~80%	N	P	0.153
Zlomke [24]	2020	Americas	64	C	>65%	30~35	60~80%	N	P	0.097

Note: T indicates that the study samples are infants and toddlers (aged 0–3). C indicates that the study samples are children between the ages of 3–6, and <50% indicates that the proportion of male children in the study sample is less than 50%. 50~65% indicates that the proportion of male children in the study sample is between 50% and 65%. >65% indicates that the proportion of male children in the study is more than 65%. <60% indicates that less than 60 percent of the mothers in the study sample had higher education. 60%~80% indicates that the proportion of mothers with higher education in the sample is between 60% and 80%. >80% more than 80% of the mothers in the study sample had higher education. N indicates that the study was conducted during a non-epidemic period. E indicates that the study was carried out during the epidemic period. I indicates that the result variable is internalizing problem behavior. E indicates that the result variable is externalizing problem behavior. P indicates that the result variable is an overall problem behavior.

## 3. Results

### 3.1. Heterogeneity Test

The heterogeneity test was carried out for the problem behaviors of preschool children, and the results were shown in Table 2. According to the results in Table 2, the *Q*-test for problem behaviors was significant, indicating that the effect sizes of the study in the meta-analysis were heterogeneous. In addition, according to the interpretation of *I*^2^ by Borenstein [26], the *I*^2^ of the meta-analysis of moral behaviors was 1452.493, which indicated that 97.01% observed variation in the relationship between maternal anxiety and preschool children’s problem behavior was caused by real differences in the relationship. *σ*^2^ represented the variance of the true effect size. *σ*^2^ showed that the true effect size had some variation. The results of the heterogeneity test showed that the random effects model we selected for the meta-analysis was accurate.

### 3.2. Publication Bias Test

First, the publication bias of this meta-analysis was examined by using a funnel plot, shown in Figure 2 for the two types of moral behavior. From the funnel plot, the research literature involving preschool children’s problem behavior was not evenly distributed on both sides of the total effect size, and most studies were located on the right side of the total effect size. This distribution suggested that there may be publication bias in the studies on preschool children’s problem behaviors. Since the funnel plot can preliminarily examine the publication bias from a subjective point of view, Rosenthal’s Failsafe N and Egger’s tests were followed to more accurately examine the publication bias, and the results are shown in Table 3.

According to the results of Egger’s test, there was a certain publication bias in the studies involving preschool children’s problem behaviors. According to Rosenthal’s N value, it is necessary to include 5 K + 10 (K is the value of effect size) in the research literature involving these behaviors to make the total effect size insignificant, indicating that there was no serious publication bias in this study.

Among the three publication bias tests mentioned above, one result (Rosenthal’s N) showed that there was no publication bias in the meta-analysis of preschool children’s problem behaviors, and the results were not recognized by all three tests. However, according to Borenstein, the purpose of the publication bias test should be to determine which of the following types of meta-analysis results belong to: 1, the effect of deviation is negligible; 2, the effect of bias cannot be ignored, but the results are still valid; and 3, the results may be problematic [26]. Therefore, further analysis was needed. We used the Trim and Fill method proposed by Duval and Tweedie to test the impact of publication bias on the results of the meta-analysis [58]. The results showed that the total effect of the random effects model on preschool children’s problem behaviors was still significant after cutting and sticking the literature. In conclusion, although there may be slight publication bias in the meta-analysis of this study, the main conclusions of the meta-analysis were still valid.

### 3.3. Main Effect Analysis

The relationship between mother’s anxiety and preschool children’s problem behaviors was examined as a whole, and the results were shown in Table 4. The results showed that there were 88 independent effect sizes of maternal anxiety and preschool children’s problem behaviors, with a total of 295,032 subjects. The overall correlation coefficient between maternal anxiety and preschool children’s problem behaviors was 0.273, as shown in Figure 3.

### 3.4. Moderation Effect Test

In Figure 3, it can be seen that the effect size of each study is distributed on the left and right sides of the total effect size (diamond in the figure), and that there is great variation among the effect sizes of each study. To analyze the variation, we respectively tested the regions (America, Asia, Europe, Australia, and Africa), age (children and infants) and gender, mother’s age, education level, and whether the research was in the epidemic period for the moderating effect of the relationship between maternal anxiety and preschool children’s problem behaviors. The results are shown in Table 5. From the results of the moderating effect analysis, the region of the subjects can affect the relationship between maternal anxiety and preschool children’s problem behaviors (*Q_b_* = 15.551, *p* = 0.004). Children’s age (*Q_b_* = 4.957, *p* = 0.026), the gender of children (*Q_b_* = 12.522, *p* = 0.002), maternal age (*Q_b_* = 11.713, *p* = 0.003), and mother’s education level (*Q_b_* = 10.124, *p* = 0.006) can also influence the relationship between maternal anxiety and preschool children’s problem behaviors.

## 4. Discussion

This study used a meta-analysis to integrate previously related research results in the field of the development of children’s problem behaviors to explore the impact of maternal anxiety on children’s problem behaviors. Specifically, we integrated previous studies on the relationship between maternal anxiety and preschool children’s problem behaviors and explored the factors influencing this relationship. The results of the meta-analysis supported the conclusion that maternal anxiety increased children’s problem behaviors [45]. Moreover, the results of the moderating effect analysis showed that the region of subjects, the age and gender of children, the mother’s age, and education level all affected the relationship between maternal anxiety and preschool children’s problem behaviors.

### 4.1. Maternal Anxiety and Preschool Children’s Problem Behaviors

In this study, a random effects model was used to conduct a meta-analysis of empirical studies on the relationship between maternal anxiety and preschool children’s problem behaviors. The results showed that there was a significant positive correlation between maternal anxiety and preschool children’s problem behaviors with an effect size of 0.273 (*k* = 88, *N* = 295,032), which was consistent with the meta-analysis of Rees [3]. Therefore, the main effect analysis results of this study proved that maternal anxiety had a significant impact on preschool children’s problem behaviors.

First of all, maternal anxiety was significantly correlated with preschool children’s problem behaviors, that is, the more preschool children were exposed to maternal anxiety, the more problem behaviors children would exhibit. Although maternal anxiety has now been widely studied as a risk factor for the development of anxiety in children, the possible role of the transmission process and genes in the association between parental and child anxiety prevalence showed that 50% of anxiety tendencies were thought to be inherited [59]. However, anxious mothers caring for their children also influenced the development of children’s anxiety [60,61], which explained why only 30% of anxiety-prone children developed anxiety disorders [61].

Second, maternal anxiety can significantly predict preschool children’s problem behaviors, that is, the more children are exposed to maternal anxiety, the more externalizing problem behaviors children exhibit. Social learning theory holds that children’s problem behaviors are learned through parenting behaviors [62,63]. And when mothers are extremely anxious, they are more likely to adopt inappropriate parenting behaviors or have a high level of control, beating and scolding. Then this process will be learned by the child, making the child’s level of problem behavior further developed. As result, preschool children unconsciously think that this kind of violence is acceptable, and may further imitate these behaviors [64], leading to externalizing behaviors such as aggressive behaviors, violent behaviors, and so on.

Finally, the more anxiety children perceive from their mothers, the more problem behaviors children will exhibit. In a growing environment, when children are frequently exposed to maternal anxiety, they may perceive that it is difficult to cope effectively with the situation, and this process will be accompanied by a strong negative emotional response. This leads to the individual adopting avoidance or destructive negative coping mechanisms and will finally lead to the emergence of problem behaviors. The cognition background theory holds that the subjective meanings of children from their mothers’ emotions and attitudes have a significant impact on their social adaptation. At the same time, some studies have proved that mothers’ emotional problems such as anxiety have a greater impact on children’s problem behaviors than parents’ attitudes [53]. Specifically, in the mothers’ anxiety environment, on the one hand, children may normalize negative emotions, and imitate and learn negative emotional expressions from their mothers, which increases the possibility of preschool children’s problem behaviors. On the other hand, when children perceive the threat from their mothers’ anxiety and inappropriate parenting behaviors, they will perceive that it is difficult for them to cope with such a negative environment, leading to maladaptive problem behaviors in children [65]. Besides, long-term exposure to negative emotions also increases the likelihood that children will develop problem behaviors.

### 4.2. Influencing Factors

#### 4.2.1. Cultural Background

The moderating effect analysis showed that the region of the subjects could influence the relationship between maternal anxiety and preschool children’s problem behaviors. First, regarding the impact of the subject’s region on the relationship between maternal anxiety and preschool children’s problem behaviors, the results showed that in the research on maternal anxiety in Asia, the average correlation coefficient between maternal anxiety and preschool children’s problem behaviors is 0.356; in the studies of America, the average correlation coefficient was 0.239; in European studies, the average correlation coefficient was 0.230; in the studies of Australia, the average correlation coefficient was 0.247; and in studies of Africa, the average correlation was 0.163. The significant moderating effect of the region of the subjects indicated that maternal anxiety was more strongly associated with preschool children’s problem behaviors in Asia than that in America, Europe, Australia, and Africa. The possible reason for this moderating effect is that the cultural background of preschool education has a certain influence on it. Transmission between different regions or different types of cultures will have an impact on children’s development and education. Therefore, it is an important basis for the operation of children’s families, childcare institutions, and communities, and the study of culture should be emphasized [66]. Second, most EU member states consider preschool education as an important part of lifelong learning strategies, and vigorously support the development of preschool education from the perspective of national policies, which has greatly reduced the pressure on mothers to raise children [67]. Finally, a large number of family support policies in European countries and America have fully alleviated the contradiction between family life and occupation, and have greatly reduced the pressure on professional women to take care of their children, thereby improving the negative emotions of mothers to prevent them from affecting the development of children [68].

The results of the moderating effect of the tested regions showed that it was more meaningful to research the internal mechanism and intervention plan of maternal anxiety affecting preschool children’s problem behaviors in Asia, which also pointed out the direction for follow-up research. The reasons are as follows: first, in terms of policy and social support, Asia does not have a relatively complete system like Europe and America. Therefore, it may be more effective to intervene in the path of maternal anxiety affecting children’s problem behaviors through other methods. Because compared with large-scale policies or the establishment of social public service systems, the promotion and implementation of small-scale intervention programs will be greatly improved. Second, children in Asia are significantly more affected by maternal anxiety than those in Europe and America, which to a certain extent affirms the theoretical and practical significance of this study or related research. Also, it can promote more effective advice on relevant educational institutions or social organizations in related fields to improve the mental health of mothers and preschool children around the world.

#### 4.2.2. Age and Gender of Children

The moderating effect analysis showed that children’s age could affect the relationship between maternal anxiety and preschool children’s problem behaviors, and the correlation coefficient between maternal anxiety and problem behaviors of children aged 3–6 was 0.333, while the correlation coefficient of children aged 0–3 was 0.270. The significant moderating effect of children’s age indicated that compared with children aged 0–3, the relationship between maternal anxiety and problem behaviors of children aged 3–6 years was stronger. The possible reason for this moderating effect is that children enter kindergarten when they are 3 to 6 years old and they begin to enter the socialization stage and contact individuals other than their mothers. Although the social influence such as classmates and friends is very limited, children’s coping styles are not perfect [69], and children aged 3–6 are still more exposed to negative emotional influences than infants and toddlers. At the same time, because mothers get more time to work when their children enter kindergarten, children aged 3 to 6 are at higher risk of exposure to negative emotions from their mothers and other sources than infants and toddlers [70]. Therefore, children aged 3–6 are more likely to be affected by maternal anxiety and exhibit more problem behaviors than children aged 0–3.

At the same time, foreign longitudinal studies have shown that the cognitive ability of preschool children to trigger their own anxiety will continue to improve with age, so the level of anxiety in various dimensions will gradually decline with the increase of children’s age [71]. However, the reason for the inconsistency of the results of this meta-analysis may be due to the following reasons: on the one hand, the problems encountered by preschool children will gradually become more frequent and complex, and the children’s expressive ability will continue to strengthen with age, so parents can more directly and accurately receive information on children’s anxiety and assess their anxiety mood. This is two-sided. When children’s cognition of dangerous stimuli deepens with age and cognitive ability, it will also lead to an increase in problem behaviors [71]. On the other hand, 3 to 6-year-old children will face the kindergarten stage, so their mothers will have stricter requirements for them. However, such strict and more diverse requirements will easily lead to children’s resistance psychology, which will easily lead to various problem behaviors [72].

Moreover, the moderating effect analysis showed that the child’s gender also affected the relationship between maternal anxiety and preschool children’s problem behaviors. And with the increase in the proportion of male subjects, the effect size between maternal anxiety and preschool children’s problem behaviors gradually decreased. This result suggested that the link between maternal anxiety and preschool children’s problem behaviors is stronger in girls than in boys. The possible reason for this moderating effect is: first, due to completely different physiological structures, boys and girls will have different psychological responses and attribution methods to the same problem [73]. Comparatively speaking, girls are more delicate in their minds and are more sensitive to dangerous situations. And they may even concretize and exaggerate dangerous stimuli, and are prone to adopt a more negative attribution method, which leads to various problem behaviors [73]. Second, based on traditional culture and social consensus, girls’ physical strength and athletic ability are relatively weak, so they are more likely to suffer physical injuries, and a more restrained personality is more likely to lead to problem behaviors such as anxiety [74].

#### 4.2.3. Mother’s Age and Education Level

Results showed that maternal age could affect the relationship between maternal anxiety and preschool children’s problem behaviors, and with the increase of maternal age, the effect size between these two gradually increased. The significant moderating effect of maternal age indicated that the relationship between anxiety states and preschool children’s problem behaviors was stronger in mothers over 30 years old than in mothers under 30 years old. The possible reason for this moderating effect is that older mothers have greater parenting pressure, and are more easily affected by their emotions when dealing with children’s various problems, making young children more prone to behavioral problems [75]. Since mothers of children under 30 are mostly new mothers, they will pay more attention to making rules for their children to avoid mistakes, will be more sensitive to children’s negative performance, and will more focus on the adverse effects of their own negative emotions on children [75].

Besides, moderating effect analysis showed that the maternal education level could also affect the relationship between maternal anxiety and preschool children’s problem behaviors, and as the proportion of mothers with higher education increased, the effect size between maternal anxiety and preschool children’s problem behaviors decreased gradually, suggesting that mothers without higher education were more strongly associated with preschool children’s problem behaviors than mothers with higher education. The possible reason is that the higher the mother’s education level, the lower the incidence of children’s behavioral problems. This may be because when mothers’ emotional problems have adverse effects on children, mothers with lower education levels cannot adopt more scientific and reasonable parenting methods to guide children to face them correctly [56].

Second, mothers with higher education levels will be more objective and accurate in their cognitive assessment of dangerous stimuli. Individuals with a higher education level will be more truly aware of dangerous stimuli, more rationally choose defense mechanisms, and will be more likely to have a positive self-evaluation of themselves [76]. And they will be more appropriate to deal with psychological stress within the individual. Moreover, they will pay more attention to and comprehensively deal with negative emotions. For example, they will learn more about psychological theories to help themselves relieve anxiety, and they will also take more diverse methods such as listening to music and doing sports to overcome anxiety, thereby reducing the impact of their anxiety on children’s development [77]. They also will pay more attention to the physical and mental health of preschool children, and face life with a more optimistic attitude. On the contrary, mothers with lower education may not be sensitive to the perception of negative emotions in preschool children, and cannot accurately and properly perceive the negative emotions of preschool children, thus affecting the development of children [78].

### 4.3. Limitations and Future Research

Limitations of the study are as follows. (1) The possible influence of the family’s socioeconomic status on the relationship between maternal anxiety and preschool children’s problem behaviors was not considered. In the field of research on maternal anxiety, family socioeconomic status is often considered an important factor. Since almost none of the original studies finally included in the meta-analysis reported the effect size of the family’s socioeconomic status, its role could not be analyzed. (2) The small sample size and uneven distribution in the moderating effect analysis can affect the results of the analysis. (3) Some of the original studies included in this meta-analysis did not report eigenvalues such as maternal age.

As an important research topic in child development psychology, it is of great theoretical and practical significance to explore the influencing factors of maternal anxiety on preschool children’s problem behaviors. Future research could focus on the following. (1) When discussing the role of rational factors in children’s problem behaviors, the role of irrational factors should also be considered, and the impact of maternal anxiety on it should be comprehensively explored from multiple perspectives. (2) Related research should also be carried out on the relationship between the measurement tools of children’s problem behaviors. (3) With the increase of age, the cognitive ability of the individual gradually matures. At different ages, the relationship between maternal anxiety and children’s problem behaviors may be different, and the strength of maternal factors and children’s cognitive factors may be different. So future research should focus on this aspect.

## 5. Conclusions

This meta-analysis found that there was a relationship between maternal anxiety and preschool children’s problem behaviors. The region of the subjects, children’s age and gender, mother’s age, and education level all affected the relationship between maternal anxiety and preschool children’s problem behaviors.

## Figures and Tables

**Figure 1 ijerph-19-11106-f001:**
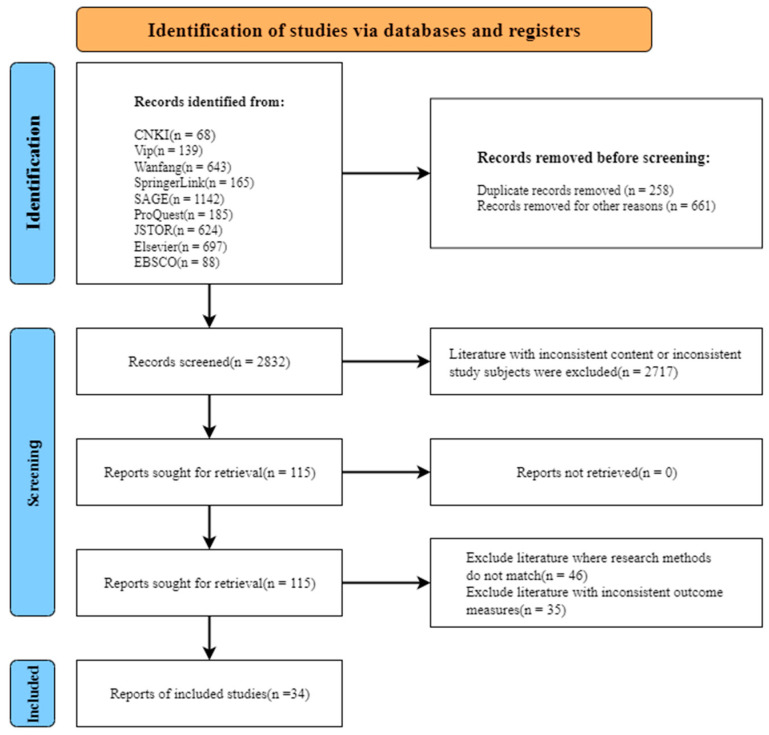
Flow chart with different phases of the search process.

**Figure 2 ijerph-19-11106-f002:**
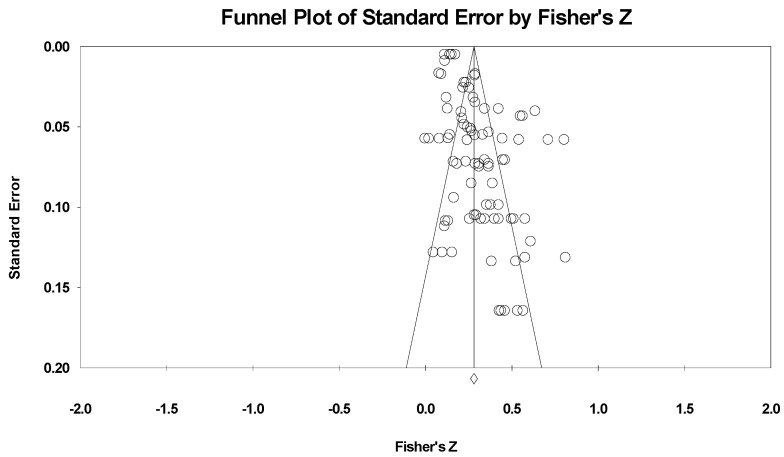
Funnel plot for research on problem behavior.

**Figure 3 ijerph-19-11106-f003:**
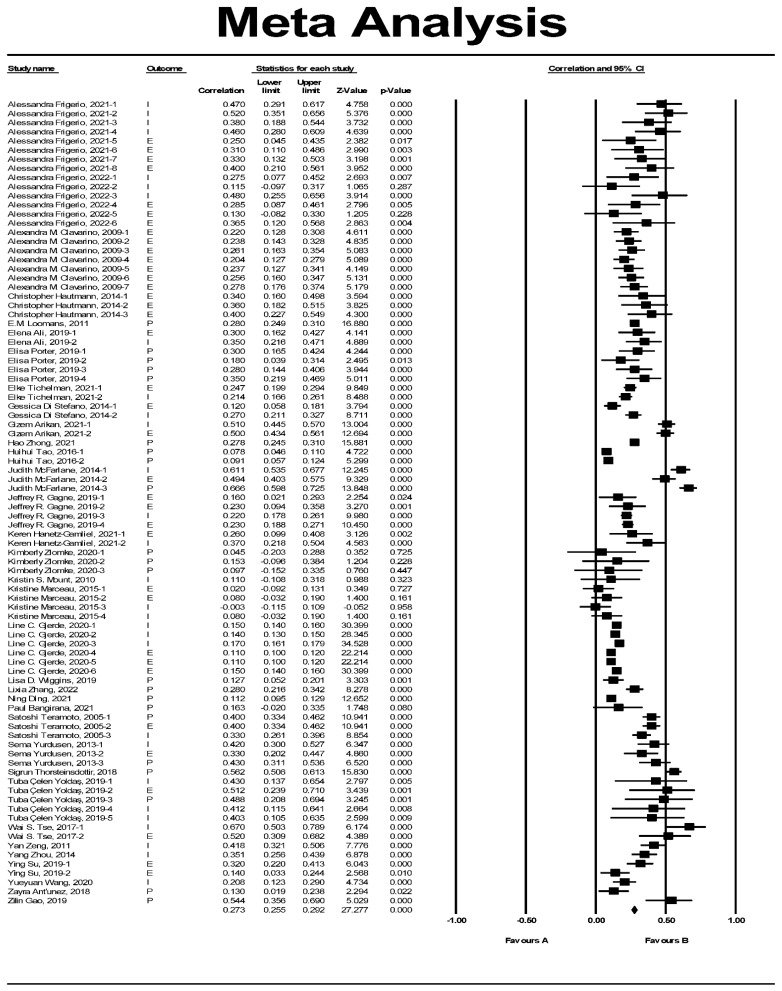
The forest map of the effect of problem behavior [2,15,22,23,24,29,30,31,32,33,34,35,36,37,38,39,40,41,42,43,44,45,46,47,48,49,50,51,52,53,54,55,56,57].

**Table 2 ijerph-19-11106-t002:** Effect heterogeneity test results.

Outcome	*Q*	*df*	*p*	*I* ^2^	*σ* ^2^
Problem Behavior	1452.493	87	<0.001	94.010	0.005

**Table 3 ijerph-19-11106-t003:** Results of publication bias test.

Outcome	Rosenthal’s N	Egger’s Intercept	*SE*	*LL*	*UL*	*p*
Problem Behavior	6737	2.94487	0.41627	2.11736	3.77238	<0.001

Note: *LL*, *UL* represents the lower limit and upper limit of 95% confidence interval of Egger’s intercept.

**Table 4 ijerph-19-11106-t004:** Analysis results of random effect model.

Outcome	*N*	*k*	*r*	*LL*	*UL*	*z*	*p*
Problem Behavior	295,032	88	0.273	0.255	0.292	27.277	<0.001

Note: *N* represents the sample size, *k* represents the number of studies, *LL* and *UL* represent the lower and upper limits of the 95% confidence interval of *r.*

**Table 5 ijerph-19-11106-t005:** Results of adjustment effect analysis.

Outcome Variable	Regulated Variable	Category	*k*	*r*	*LL*	*UL*	*Q_b_*	*p*
Problem Behavior	Region	Americas	21	0.239	0.167	0.309	15.551	0.004
	Asia	27	0.356	0.297	0.411		
	Europe	28	0.230	0.205	0.254		
	Australia	11	0.247	0.205	0.254		
	Africa	1	0.163	-0.020	0.335		
Age of Children	Child	31	0.333	0.282	0.382	4.957	0.026
	Toddler	42	0.270	0.246	0.293		
Gender (Male%)	<50%	15	0.398	0.303	0.484	12.522	0.002
	50~65%	50	0.220	0.201	0.239		
	>65%	9	0.250	0.135	0.358		
Maternal Age	>35	7	0.340	0.173	0.488	11.713	0.003
	30~35	48	0.348	0.308	0.387		
	<30	6	0.253	0.216	0.291		
Maternal Education	<60%	17	0.285	0.219	0.348	10.124	0.006
	60~80%	17	0.334	0.265	0.400		
	>80%	16	0.225	0.199	0.250		
Period	Non-epidemic period	79	0.272	0.253	0.291	0.081	0.776
	Epidemic period	9	0.280	0.229	0.330

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
