# Peer review of "Association between Maternal Anxiety and Children’s Problem Behaviors: A Systematic Review and Meta-Analysis"

_ijerph, 2022, doi:10.3390/ijerph191711106_

Round 1

Reviewer 1 Report

The study is a meta-analysis of the relationship between maternal anxiety and children behavioural difficulties. By the analysis of 88 studies, it is suggested that beside a significant relationship between maternal anxiety and children behavioural difficulties, there are many differences across studies and geographical region and children gender.

Although my low expertise in reviewing met/analysis, the method of the study seems well done and the results can contribute to the existing literature. Some minor observations follow:

-      In the abstract and in the keywords the target population misses (preschoolers)

-       In the introduction I suggest referring these sentences “Studies have found that there is a correlation between mothers’ emotions and children’s emotions. However, previous studies have mainly explored the causes of mothers’ anxiety, mothers’ own mental health status and the direct or indirect effects of changes in parenting styles on children. “

-       This period is not clear to me “Results of a  long-term follow-up study showed maternal anxiety at children’s 3 months, 6 months, and 12 months and at 10 and 12.5 years of age. Mother-reported child internalizing problem behaviors were significantly associated with maternal anxiety, but not for children of externalizing problem behaviors, but were not significantly associated with maternal anxiety or depressive symptoms (Missler, Straten, Denissen, Donker, Weerth, & Beijers, 2021). Pag 4, lines 144-149

-       I suggest discussing within the introduction the results of the systematic meta-analysis of Rees, 2018 as it is the starting point of this work. I also suggest expanding the reason why a new meta-analysis is needed. Pag.4, lines 153-155.

-       The search stringa used does not include the keyword “preschool*” which is the target of the study, as indicated in the study aims and included in the inclusion criteria. It is true that "children" can include preschool population, however, some articles may use only the term “preschoolers” and they could be missed in the databases research.

-       It is not clear if the authors put a time range for the search of the study. It is important also because results could be compared to the previous systematic meta-analysis

-       I suggest revising some formal aspects throughout the manuscript, such as upper letter or spaces in the correct positions. I underline some of these:

·      “…in preschool is period”, please delete “is” (pag. 3, line 118)

·      on defined predefined criteria” pag 5, line 188

·      Please delete “that” from “For example, a study by Zlomke (2020) found that no correlation between maternal anxiety and children’s externalizing problem behaviors” pag 4 lines 151-153

·      In the following sentence the subject misses “although maternal anxiety has now been widely studied as a risk factor for the development of anxiety in children, and the possible role of transmission process and genes in the association between parental and child anxiety prevalence (Bogels & Brechman-Toussaint, 2006) shows that 50% of anxiety tendencies were thought to be inherited. “

Author Response

Dear Reviewer 1, we have carefully read your suggestion and revised the manuscript according to your suggestion. We have responded to your suggestions one by one. Please see the attached file titled "Response to Reviewer 1 Comments".

Reviewer 2 Report

The paper is focused on an important issue. It could be improved by focusing the literature on referring to previous studies and published definitions of maternal anxiety and how it is measured and when e.g., prenatal and postnatal. Also reference to how maternal anxiety relates to the development of primary caregiver/mother secure attachment relationships would be helpful. Different measures of maternal anxiety appear to be used in the studies included in the meta-analysis e.g., Prenatal anxiety (Ali et al 2020) and Brief Symptom Inventory (Arikan & Kumru, 2020). It would be helpful to explain these different measures and any impact these may have on the meta-analysis. Also where studies have focused on children at high risk of autism compared to typical developing infants, the measures reflect this focus (e.g., Broad Autism Phenotype Questionnaire) and the measure of anxiety is the Parenting Stress Index Form (Ding et al, 2021). Inclusion criteria for the meta-analysis focused on pre-school children, however it is not clear if pre-natal anxiety and post-natal anxiety and also pre and post natal depression are controlled for in the analysis. The discussion states that "maternal anxiety positively affected children's problem behaviours" (line 326). However this statement seems misleading. Also the results of moderating effect analysis is stated to how that the region of subjects also affected the relationship between maternal anxiety and preschool children's problem behaviours. However the region is determined by the continent of the study as opposed to a more refined coding of subjects' cultural background and ethnicity or level of deprivation and socio-economic status. Overall the discussion seems to present a number of assertions without reference to published evidence or theoretical background e.g., line 444 -449; line 458-464; line 471-477; 485-487; 489-502. Section 4.1 could refer to attachment relationships. Section 4.2.1 seems to make based on assumptions that each study is a cultural reflection of their continent and there is only one African study included in the meta-analysis. 

Author Response

Dear Reviewer 2, we have carefully read your suggestion and revised the manuscript according to your suggestion. We have responded to your suggestions one by one. Please see the attached file titled "Response to Reviewer 2 Comments".

Reviewer 3 Report

Dear authors,

The question your study attempts to answer is important and significant for contemporary children's mental health. I was looking forward to discovering your findings. 

However, the paper, in its current form is very difficult to read and even more difficult to follow as there are numerous problems related to English use. Besides, the references are not numbered in the text and are almost impossible to verify. As a result, I recommend a thorough rewriting and English checking with appropriate referencing.

Author Response

Dear Reviewer 3
Thank you very much for your generous suggestions. We have checked and improved the entire article for the use of English. For details, please refer to our latest uploaded manuscript. In addition, we have numbered references. We are very sorry for the inconvenience caused to you in reading the article.
We hope you find our revised manuscript improved and suitable for publication in IJERPH.

Round 2

Reviewer 3 Report

Dear authors, 

I appreciate the fact that you went seriously over the study and made it much easier to read and understand what is a serious amount of work on an important topic. I also agree with the limitations and future research needed on the matter.